# Divergent Contribution of the Golgi Apparatus to Microtubule Organization in Related Cell Lines

**DOI:** 10.3390/ijms232416178

**Published:** 2022-12-19

**Authors:** Ilya B. Brodsky, Artem I. Fokin, Aleksei A. Efremov, Elena S. Nadezhdina, Anton V. Burakov

**Affiliations:** 1A.N. Belozersky Institute of Physico-Chemical Biology, Lomonosov Moscow State University, Moscow 119992, Russia; 2CNRS UMR168, Institut Curie, 75005 Paris, France; 3Faculty of Bioengineering and Bioinformatics, Lomonosov Moscow State University, Moscow 119234, Russia; 4Institute of Protein Research of Russian Academy of Sciences, Moscow Region, Pushchino 142290, Russia

**Keywords:** microtubules (MT), Golgi, centrosome, microtubule organization, transcriptome analysis

## Abstract

Membrane trafficking in interphase animal cells is accomplished mostly along the microtubules. Microtubules are often organized radially by the microtubule-organizing center to coordinate intracellular transport. Along with the centrosome, the Golgi often serves as a microtubule-organizing center, capable of nucleating and retaining microtubules. Recent studies revealed the role of a special subset of Golgi-derived microtubules, which facilitates vesicular traffic from this central transport hub of the cell. However, proteins essential for microtubule organization onto the Golgi might be differentially expressed in different cell lines, while many potential participants remain undiscovered. In the current work, we analyzed the involvement of the Golgi complex in microtubule organization in related cell lines. We studied two cell lines, both originating from green monkey renal epithelium, and found that they relied either on the centrosome or on the Golgi as a main microtubule-organizing center. We demonstrated that the difference in their Golgi microtubule-organizing activity was not associated with the well-studied proteins, such as CAMSAP3, CLASP2, GCC185, and GMAP210, but revealed several potential candidates involved in this process.

## 1. Introduction

Membrane trafficking along microtubules (MTs) plays an essential role in a variety of intracellular processes. Microtubule system architecture coordinates intracellular transport and determines its preferred direction and intensity. Microtubules, therefore, are often organized in a radial array that ensures efficient transport of cellular components [1]. In many cell types, the centrosome serves as a dominant microtubule-organizing center (MTOC), featuring microtubule-nucleating and -anchoring activities [2]. However, in addition to the centrosome, the spots of microtubule nucleation can also be presented on the Golgi membranes [3,4], the nuclear envelope [5,6], the cell cortex [7], and some other places, such as chromatin, kinetochores, or pre-existing MTs (reviewed in [8]). It is well known that Golgi, along with the centrosome, serves as a potent MTOC, capable of nucleating MTs and retaining them in an organized fashion. The centrosomal MTs might complement the activity of the Golgi-derived ones, which in turn ought to adjust the proper self-assembly of the Golgi [9]. In some mammalian cells, such as retinal pigment epithelium cells, RPE1, approximately half of all microtubule nucleation events are observed at the Golgi membranes [10]. To date, Golgi-derived MTs have been discovered in hepatocytes, epithelial cells, neurons, skeletal muscles, and pancreatic beta cells, where this MT subpopulation finetunes insulin secretion (reviewed in [11]). 

The spatial overlap of the Golgi and the microtubule convergence center, as well as the colocalization of dispersed Golgi vesicles and individual microtubules, was observed for the first time almost 40 years ago [12,13]. Later, it was shown that the Golgi itself was able to support microtubule nucleation both in vitro and in vivo [3]. The Golgi was found to nucleate new microtubules in specialized cell types, such as polarized Drosophila neurons, mouse muscle cells, and rodent oligodendrocytes. Thus, the cells with specialized shapes and functions are thought to rely on Golgi outposts for local microtubule nucleation and organization [14].

Microtubule organization onto the Golgi requires the coordinated work of many proteins. The initial role in this process is assigned to the γ-TuRCs (γ-Tubulin Ring Complexes) [3], which are responsible for the nucleation of microtubules throughout the whole cell [15]. γ-TuRCs bind to the Golgi membranes through adapter proteins, one of which is GMAP210 [16]. GM130 at the cis-Golgi binds AKAP450, which in turn links CEP215 and MMG to recruit γ-TuRCs [17]. After being released from γ-TuRCs, microtubules are stabilized by CAMSAP2 and CLASP1/2, which are connected to the trans-Golgi through GCC185 [4]. GCC185 is recruited to the Golgi membranes through its short GRIP domain located at the C-terminus of the molecule. CLASP2 binds to GCC185 with its C-terminus, while the free N-terminals of the resulting CLASP2 dimers capture α-β-tubulin and thus ensure the nucleation and anchoring of microtubules at the Golgi membranes [18]. Another protein, MTCL2, associates with the Golgi membrane through the N-terminal coiled-coil region and directly binds microtubules through the conserved C-terminal domain without promoting microtubule stabilization [19]. Moreover, the interaction between MTCL1 and CLASP1/2 can tether microtubules, whereas the length of the microtubules is regulated by CAMSAP2 when it associates with EBs proteins. The GM130 interactor GRASP65 regulates microtubule stabilization. Thus, microtubule organization on the Golgi is orchestrated by a complex hierarchy of protein interactions that ensure the nucleation and anchoring of microtubules [17,20]. It should be noted that the structural integrity of Golgi is also a prerequisite for its work as an MTOC. An example is the work of the Golgi-associated protein GRASP65, which is well known for its role in Golgi stacking and ribbon formation. This protein is not involved in microtubule nucleation or anchoring, but it is required for the stabilization of newly nucleated microtubules, leading to their acetylation and clustering of Golgi stacks, thus required for the organization of the microtubule cytoskeleton [21]. 

The Golgi surface is very heterogeneous and contains discrete and sparse hotspots that facilitate clustered microtubule nucleation [22]. This indicates that the Golgi functioning as a microtubule-organizing center is a complex process that is precisely controlled in the cell based on its current needs. Recent studies have revealed the role of a special subset of Golgi-derived MTs, which serve as fast tracks for the anterograde intracellular transport that ensures persistent cell migration [23]. Thus, the study of MT organization onto the Golgi could also lead to the understanding of how fast membrane trafficking is coordinated in the cytoplasm.

Despite deciphering key players involved in microtubule organization onto the Golgi membranes, many potential partners remain undiscovered. In addition, among the characterized ones, some might be differentially expressed in different cell lines. In the current work, we analyzed the involvement of the Golgi complex in microtubule organization in related cell lines. We found two related cell lines originated from green monkey renal epithelium, which relied either on the centrosome or on the Golgi as a main MTOC. This phenomenon was not directly connected to the expression level of well-studied proteins, such as CLASP2, GCC185, or GMAP210. Further transcriptome analysis revealed a difference in the expression levels of several genes associated with both microtubules and the Golgi. The corresponding proteins could be potentially involved in the microtubule organization by the Golgi and thus facilitate vesicular traffic towards this central transport hub of the cell. 

## 2. Results

### 2.1. Morphological Analysis of the Microtubule System Revealed a Difference in Microtubule Architecture between the Two Lines of Green Monkey Kidney Cells 

The microtubule-organizing activity of the Golgi may vary greatly in different cells. Classical concepts about the microtubule architecture were obtained using common cell lines such as retinal pigment epithelium or fish melanophores. These cell lines differ in many parameters, which renders a correct comparison impossible. We chose two epithelial cell lines, Vero and BS-C-1, derived from the kidney of a green monkey (Cercopithecus aethiops), which had similar morphology. One of them, BS-C-1, had already been shown to be able to nucleate microtubules at the Golgi vesicles [24]. 

To understand where in the cell the microtubule network is centered, we labeled microtubules, the Golgi, and the centrosome in both the BS-C-1 and Vero cells (Figure 1A,B). Most of the cells displayed a radial microtubule aster focused on the cell center, where the Golgi apparatus encompassed the centrosome (Figure 1C). Microtubules could also converge to a wide central area, where the Golgi presumably had an impact on microtubule organization. Some cells possessed two independent MTOCs; in this case, the Golgi appeared as a typical cap-like shape and was placed apart from the centrosome (21.3% in BS-C-1 and 8.7% in Vero). At times, it was obvious that the microtubules were associated only with the centrosome (1.2% in BS-C-1 and 4.8% in Vero) and vice versa with the Golgi but not with the centrosome. (7.2% in BS-C-1 and 2.1% in Vero). In a small fraction of cells, the microtubule system appeared disorganized (22.2% of the Vero and in 16.6% of the BS-C-1 cells (Figure 1C). 

To summarize, our analysis showed that both MTOCs were involved in microtubule organization in both cell lines, with a slight predominance of the centrosome in the Vero cells and the Golgi in the BS-C-1 as the main MTOCs.

### 2.2. The Centrosome and the Golgi Have Different Impacts on Microtubule Organization in the Vero and BS-C-1 Cells

To confirm our hypothesis regarding the differential contribution of the centrosome and the Golgi to microtubule organization in Vero and BS-C-1, we disrupted each of these MTOCs separately. A direct way to disrupt the Golgi is to treat cells with Brefeldin A, which blocks vesicular transport from the endoplasmic reticulum to the Golgi, thus leading to the collapse of the latter. Generally, the microtubule system appeared more damaged in the BS-C-1 cells than in the Vero, where most of the cells retained a radial microtubule array after BFA treatment. (Figure 2A,C and Appendix A) and even appeared more radial than in the control (Figure 2C).

To eliminate the centrosomes from cells, we first treated them with centrinone B, which causes reversible centriole depletion by inhibiting Polo-like kinase 4 [25]. We tested the effect of the centrinone B treatment by adding it to the culture medium at a 500 nM concentration and performed the immunostaining of cells for γ-tubulin and α-tubulin a week later (Figure 2B,C, and Appendix A). We found that the microtubule radiality degenerated to a greater extent in the Vero cells than in the BS-C-1. In fact, there were almost no Vero cells containing any sort of ordered microtubules (Figure 2C and Appendix A). 

We then mechanically removed nuclei from the Vero and BS-C-1 cells by centrifugation and obtained a mixed population of cytoplasts containing or lacking centrosomes, which could be easily distinguished by the gamma-tubulin immunostaining. (Figure 2E). The cytoplasts containing centrosomes obtained from both cell lines demonstrated radial microtubule aster, which looked similar to those in intact cells (Figure 2E). The enucleated BS-C-1 cells lacking centrosomes looked similar, and their microtubules diverged from the cell center to the periphery, as was shown earlier [26,27]. Acentrosomal cytoplasts obtained from the Vero cells, in contrast, revealed a chaotic distribution of microtubules, which indicated the determining role of the centrosome in the Vero cells (Figure 2E,F). This was consistent with the previous results showing that the relative impact of the centrosome on microtubule organization was higher in Vero cells. Thus, under centrosome depletion, the situation was the opposite of that when the integrity of the Golgi was compromised using Brefeldin A treatment. Taken together, these results confirmed our initial suggestion that the Golgi participated in microtubule organization to a higher degree in the BC-S-1 than in the Vero cells.

### 2.3. Different Nucleation Activity at the Golgi Membranes in the Vero and BS-C-1 Cells Coexists with the Same Activity of Their Centrosomes

We monitored the early stages of microtubule regrowth in the nocodazole washout experiments. Cells were treated with nocodazole and cooled on ice to fully depolymerize the remaining microtubules. Nocodazole was washed out for 4 min in a warm medium. The cells were then fixed and immunostained to visualize the microtubules and the Golgi (Figure 3A). Dispersed Golgi vesicles were presented in the cytoplasm as GM-130-positive dots. In each cell, an aster formed by short microtubules extending from the centrosome was clearly visible; in addition, many short non-centrosomal microtubules arose in the cytoplasm. (Figure 3B). We found that approximately three times more newly nucleated cytoplasmic microtubules were associated with GM-130-positive vesicles in the BS-C-1 cells than in the Vero (46.55% versus 14.68%). This allowed us to conclude that the microtubule-nucleating activity of the Golgi was indeed higher in the BS-C-1 versus Vero (Figure 3B). The activity of the alternative MTOC (Golgi) could compensate for the low activity of the primary MTOC (the centrosome) if the pool of microtubule-nucleating factors in the cell is restricted and must be redistributed between the centrosome and the Golgi [11]. To find out whether the difference in the Golgi nucleating activity was due to compensatory mechanisms resulting from the centrosome dysfunction, we decided to compare the nucleation activity of the centrosomes in Vero and BS-C-1 cells.

In the nocodazole washout experiments, it was not possible to accurately quantify the number of microtubules nucleated at the centrosome, due to their high density. Therefore, we proceeded to in vivo fluorescence microscopy. We expressed GFP-fused microtubule plus-end marker EB-3 in cells; thus, we could visualize nucleation events at the centrosome and/or the Golgi by observing where the GFP-EB3 comets flew from (Appendix A and Figure 3C) as it was performed earlier [4]. Using this approach, we counted the rate of microtubule nucleation at the centrosome (see Materials and Methods). Then, we fixed the cells and immunostained them with the antibodies to mannosidase II and dynactin subunit p150^Glued^ to determine the precise localization of the centrosome and the Golgi (Figure 3E). Thus, we could accurately evaluate the nucleating activity of the centrosome by counting the EB3 comets arising from it and compare it in the Vero and BS-C-1 cells (Figure 3D). 

Analysis of the movies revealed that the nucleation activity of the centrosomes was equal in both cell lines. On average, we observed 17 microtubule nucleation events per minute at the centrosomes in the Vero cells and 18 in the BS-C-1 cells. (Figure 3D). Thus, the hypothesis that increased microtubule-nucleating Golgi activity in the BS-C-1 cells was due to the redistribution of nucleating factors from the centrosome to the Golgi membranes was not confirmed. Therefore, we verified that in both related cell lines, the centrosomes were equally active, but the Golgi membranes for some reason nucleated microtubules much better in the BS-C-1 than in the Vero.

### 2.4. CAMSAP3 Overexpression Alters Microtubule Organization in the Golgi-Independent Manner

Rather few (~15–20) proteins were shown to participate in the process of nucleation and the anchoring of microtubules onto the Golgi [7,8,10,11,17]. We decided to investigate which of them may be responsible for the difference in the Golgi MT-organizing activity between the Vero and BS-C-1. To assess the expression of the Golgi and microtubule-associated proteins in these cell lines, we performed a transcriptomic analysis. The experiment was repeated twice, with a high level of convergence of results. A total of 35,300 transcripts were identified in both cultures, of which 24,433 belonged to the identified genes (NCBI Sequence Read Archive project id PRJNA862476).

To our surprise, proteins previously noted as participants in the assembly of microtubules on the Golgi apparatus (GOLGA2 (GM130), AKAP9 (AKAP450), GCC2 (GCC185), CAMSAP1/2, CLASP1/2, CLASRP, CEP350, GORASP1 (GRASP65), GMAP210 (TRIP11), MTCL1, CDK5RAP1/2/3, and ninein (NIN)) were expressed in the Vero and BS-C-1 at approximately the same levels for several hundred reads each (Table 1). Myomegalins (MMG) were some exceptions. In the BS-C-1 and Vero, they were represented by the LOC103247162, LOC103224073, LOC103224075, and PDE4DIP transcripts, which were expressed at a close to zero level in both cultures and were not found in the general transcriptomic list. On the whole, the study of transcriptomes showed almost no differences in the expression level of proteins that ensure the assembly of microtubules on the Golgi apparatus in the studied cultures. CAMSAP3 was an exception as its level was 3.75 times lower in Vero than in BS-C-1 cells. However, to date, only the role of its paralog, CAMSAP2, in microtubule organization at the Golgi was demonstrated [17,20]. Nevertheless, we decided to investigate CAMSAP3, since as shown by mRNA sequencing, among the MT and Golgi-associated proteins, only CAMSAP3 was expressed differentially in the studied cell lines. 

In the previous study, Wang with coauthors demonstrated that this protein could be important for the organization of non-centrosomal microtubules as well as for the assembly of the Golgi apparatus in a centrosome-independent manner [28]. To determine whether the level of CAMSAP3 could be the determining factor for the Golgi capacity to organize the microtubule, we decided to increase the level of CAMSAP3 in Vero cells. We transfected the Vero cells with the plasmid-encoding CAMSAP3-GFP and studied the microtubules and the Golgi. We found that even the lowest levels of expression lead to total disruption of microtubule radiality, while high levels cause microtubule bundling; the Golgi is usually fragmented in transfected cells (Figure 4A).

In the microtubule regrowth experiments, we found that, in addition to nocodazole-resistant microtubule bundles, CAMSAP3 also induced the formation of numerous non-centrosomal microtubules (Figure 4B). However, no visible colocalization between these microtubules and dispersed Golgi vesicles was found as shown by wide-field and confocal microscopy (Appendix A). Another intriguing observation was that CAMSAP3 overexpression exhausted the cytoplasmic pool of soluble tubulin, which was relocated to the bundled microtubules (Appendix A).

Thus, we could conclude that CAMSAP3 is involved in non-centrosomal microtubule nucleation, which apparently takes place independently of the Golgi membranes. As a very potent MT-nucleating protein, whose role in non-centrosomal MT nucleation was previously demonstrated [28,29], CAMSAP3 might influence the Golgi morphology indirectly through the regulation of the MT architecture.

### 2.5. The GCC185-CLASP2 Axis Is Not Determining for the Golgi Activity as an MTOC

Although the transcriptome analysis showed no difference in the expression level of well-known proteins that ensure the assembly of microtubules on the Golgi, such proteins could be a subject of proteasomal degradation and many other post-translational modifications. Thus, we decided to study the role of the best-characterized proteins such as CLASP2, whose role in MT organization on the Golgi has been supported by numerous publications (reviewed in [8,11]). 

We found that the level of CLASP2 was elevated in the BS-C-1 cells compared to the Vero (Figure 5A,B). GMAP 210 was poorly detected in cell lysates, and immunofluorescent staining revealed no morphological differences between the BS-C-1 and Vero cells (Appendix A). Our study, therefore, was focused on CLASP2 and GCC185 as its intermediary. Localization of endogenous CLASP2 did not appear to be different between the two cell lines (Figure 5C): The endogenous protein was distributed throughout the cytoplasm, with some affinity for the central region where the microtubules converged. The next step was to overexpress exogenous CLASP2 in the Vero cells to reawaken the Golgi microtubule-organizing activity. GFP-fused CLASP2 overexpression in the Vero cells resulted in most of the exogenous protein being distributed along the microtubules. GFP-CLASP2 also partially co-localized with the mannosidase-II-positive vesicles in the central region of the cell (Figure 5D). To study whether the exogenous CLASP2 could directly affect the microtubule nucleation at the Golgi membranes, we conducted microtubule regrowth experiments after the nocodazole washout in the GFP-CLASP2-transfected Vero cells. Indeed, we observed numerous short microtubules decorated with GFP-CLASP2 in the cytoplasm of transfected cells (Figure 5E, red arrows). In addition, GFP-CLASP2 also decorated microtubules at the centrosome aster (Figure 5E, green arrows). Visually, the transfected cells contained more cytoplasmic microtubules, which were mainly CLASP2-decorated. However, they might have remained in the cytoplasm due to the higher resistance to nocodazole, since CLASP2 strongly stabilizes microtubules. Indeed, having many short cytoplasmic microtubules was not associated with mannosidase II-positive vesicles (Figure 5E), and the number of Golgi-associated cytoplasmic microtubules was still lower than that observed in the BS-C-1.

To enhance the binding of exogenous CLASP2 to the Golgi surface, we created a chimeric construct GFP-CLASP2-GRIP. It was previously shown that the GRIP domain was able to target GFP or even larger proteins on the surface of the Golgi apparatus [1,30,31]. In the microtubule regrowth experiments, we observed that GFP-CLASP-GRIP did colocalize to dispersed Golgi vesicles (Figure 5F). However, many cytoplasmic microtubules decorated with exogenous CLASP2 still originated from the foci where no mannosidase staining was observed.

Since CLASP2 does not directly bind to the Golgi membranes but uses Golgin GCC185 as an intermediary, we decided to try to express GCC185 in the Vero cells. We conducted a series of microtubule recovery experiments in the Vero cells expressing mCherry-GCC185. Exogenous GCC185, along with a homogeneous distribution throughout the cytoplasm, decorated numerous vesicles, which partially coincided with mannosidase2-positive dots. This nonetheless decreased overall levels of microtubule nucleation in transfected cells, both in the cytoplasm and at the centrosome (Figure 5G). 

These results suggest that the GCC185-CLASP2 axis is not fully responsible for the microtubule nucleation at the Golgi in our cell system. None of its components, when overexpressed, could reinforce the weak microtubule-organizing capacity of the Golgi in the Vero cells to the level of the one in the BS-C-1 cells. Eventually, one could suggest the existence of one or more unknown participants in the process of microtubule nucleation at the Golgi membranes. To explore this possibility, we decided to deeply analyze our transcriptomic data to find novel regulators of the MT organization onto the Golgi.

### 2.6. Transcriptomic Analysis Reveals Several Hitherto Unknown Potential Factors That May Influence the Golgi Functioning as a Microtubule Organizer

We found that the transcriptome composition of the cultures differed markedly. It turned out that 63 genes were expressed two orders of magnitude higher in the BS-C than in the Vero, and 52 genes were expressed in the Vero 100 times higher than in the BS-C-1. A total of 4163 genes were predominantly (>2) expressed in the BS-C-1 than in the Vero, and 2945 genes were predominantly (>2) expressed in the Vero. Thus, approximately 29% of the identified genes differed significantly in their expression levels in the two studied green monkey kidney cultures. Hereafter, we will denote the ratio of the expression level in the BS-C-1 to the expression level in the Vero as B/V = X.

The genes expressed differentially in these cell lines belonged to very different functional groups. Consequently, we focused only on the genes related to microtubule assembly on the Golgi apparatus. Typically, most of them were assumed to have invariant expression levels. 

First, we analyzed microtubule and centrosome proteins. In the Human Protein Atlas database, these proteins were represented by 1134 names, among which were microtubule structural proteins, centrosome proteins, microtubule motor proteins, and microtubule interacting proteins. We did not find 83 proteins in our transcriptomic list, and 43 genes showed zero or close to zero expression levels. Of the remaining genes, 127 were predominantly present in the BSC-1 and 91 in the Vero (Appendix A). We noticed that the second isoform of gamma-tubulin (TUBG2) was highly expressed in the Vero, though this isoform usually presents in embryonic tissues. It was also shown that γ-tubulin-2 accumulates in the adult brain. γ-tubulin-2 accumulation in mature neurons and neuroblastoma cells during oxidative stress may denote its prosurvival role in neural cells [32]. The first gamma-tubulin isoform TUBG1 and another γ-TuRC subunit, TUBGCP, were expressed at the same level in both cultures. All other tubulins were either presented at approximately the same levels or were completely absent in both cultures. 

Most surprisingly, a significant amount of MAP2, a well-known microtubule stabilizer, was present in BS-C-1 cells (B/V = 7.12) (Appendix A). This culture also actively expressed the microtubule-stabilizing protein Furry FRY (B/V = 9.55), as well as the BCAS3 microtubule-associated cell migration factor (B/V = 3.45), Tubulin polymerization-promoting protein (TPPP) (B/V = 3.5), and Doublecortin domain containing 2 (DCDC2) (B/V = 3.46). Thus, the microtubules in the BS-C-1 might be more stable than in the Vero, but this needs further investigation. In our experiments, we did not observe any difference in the sensitivity of the microtubules to nocodazole and in their growth rate upon washing. Microtubule-associated protein 1 light chain 3 gamma (MAP1LC3C) (B/V = 14.17) and Microtubule-associated protein 1 light chain 3 alpha (MAP1LC3A) were also expressed at a higher level in BS-C-1(B/V = 3.15), but both proteins were related to autophagocytosis processes rather than to microtubule regulation. In BS-C-1, the expression of the CAP-Gly domain containing linker protein family member 4 (CLIP4) was also upregulated (B/V = 3.43).

In both cultures, the centrosome proteins CEP126, CEP76, CEP295NL, and centrin-1 were not expressed or were not found in the transcriptomic data. In the BS-C-1, the expression of the centrosomal protein 85-like protein (CEP85L) was more pronounced (B/V = 3.58). Other centrosomal proteins were expressed at approximately the same levels. Microtubule stability could also be regulated by kinesins. Both cultures expressed 43 kinesin genes. KIF2B was not found in the screening results, and KIF19 was expressed at a low level only in the BS-C-1. KIF5C, KIF3C, KIF6, KIF26B, and KIF21B were upregulated in the BSC-1, while KIF5A and KIF25 were upregulated in the Vero (Appendix A). The kinesin-binding protein MAP7 domain containing 2 (MAP7D2) was somewhat prevalent in the Vero (B/V = 0.33).

Dyneins, dynactins, and accessory proteins were expressed at approximately the same level in both cultures. The Dynein cytoplasmic 1 intermediate chain 1 was somewhat prevalent in the BS-C-1 (B/V = 2.65). Other exceptions also included the Dynein axonemal intermediate chain 2 (B/V = 4.25), Dynein light chain Tctex-type 2 (B/V = 3), Dynein light chain roadblock-type 2 (B/V = 2.71), and the Dynein axonemal heavy chain 3 (B/V = 2.24).

In total, 919 proteins could be attributed to the Golgi apparatus (Human Protein Atlas database), of which 194 were expressed at the zero level or were absent in our transcriptome lists (Appendix A). A total of 177 genes were upregulated (>2) in the BS-C-1 and 106 genes in the Vero. Among the proteins of the Golgi apparatus interacting with microtubules and more represented in the BS-C-1, we noted Janus kinase and microtubule interacting protein 2 (JAKMIP2) (B/V = 3.75), Janus kinase and microtubule interacting protein 3 (JAKMIP3) (B/V = 2.08), and Microtubule-associated monooxygenase, calponin, and LIM domain containing 2 (MICAL2) (B/V = 4.63). The protein GCC1 (GCC88) (B/V = 3.46) interacts with GCC185 and actin and participates in the formation of the trans-Golgi network and endosome transport, i.e., it could also be a regulator of microtubule assembly. FAM198b (GASK-1B) was 220 times upregulated in the BS-C-1. This protein is involved in the regulation of matrix metalloproteinase secretion, and its interaction with microtubules cannot be ruled out. The Vero cells express fewer laminins (3.29 times in total) and collagens (1.77 times) than the BS-C-1; therefore, these cells might possess less “powerful” Golgi. Therefore, they might need fewer Golgi-derived microtubules as fast tracks for cargo transport.

Finally, we decided to consider the GTPases that were actively involved in the functioning of the Golgi apparatus separately (Appendix A). We considered the Arl (26 genes) and Rab (85 genes) families. Of these, Arl4c prevailed in the BS-C-1 (B/V = 4.62), and Arl11 prevailed in the Vero (B/V = 0.018). Rab36 was expressed only in the BS-C-1. Rab20, Rab22a, and Rac2 were upregulated in the BS-C-1 (B/V = 7.06, 3.16, and 3.36, respectively), Rab3b and Rab7b were upregulated in the Vero (B/V = 0.28 and 0.03, respectively). Rab11FIP4 was somewhat larger in the BS-C-1 (B/V = 3.1).

Thus, our comparison of the transcriptomes of the Vero and BS-C-1 cells demonstrated, on the one hand, the similarity in the level of gene expression of almost all proteins, for which their role in the organization of microtubules at the Golgi membranes was shown. At the same time, significant differences were found in the level of expression of several genes encoding poorly studied proteins that could contribute to these processes.

## 3. Discussion

In the current work, we studied how the central transport hub of the cell, the Golgi, organizes its access roads for vesicular traffic, i.e., acts as a microtubule-organizing center. We described, for the first time, a significant difference in Golgi microtubule-organizing capacity between two cell lines of apparently related origin. Specifically, we have shown that the removal of the centrosome, either mechanically or by exposure to centrinone B, had a stronger effect on the microtubule system in the Vero cells. At the same time, the BS-C-1 cells were not affected by centrosome depletion, likely due to the compensatory mechanisms provided by an alternative MTOC, the Golgi. In contrast, microtubules were more affected by the Golgi dispersion with Brefeldin A in BS-C-1. This correlates well with the previous finding, showing that the BS-C-1 cytoplasts were unable to organize microtubules in the absence of the Golgi [33] but successfully organized a microtubule aster without the centrosome [26,27]. This is also consistent with previous studies in which inhibiting microtubule nucleation at the Golgi does not affect centrosomal activity, whereas removing centrosomes modifies the microtubule nucleation activity of the Golgi apparatus [34].

The general analysis of the microtubule system in connection with the two MTOCs, as well as the ablation of both separately, provided insight into the cell preference towards one particular MTOC. However, only one of the two main features of an MTOC, the capacity to anchor microtubules, was analyzed there. To study nucleation, we conducted a series of microtubule regrowth experiments and a series of direct observations of EB3-GFP “comets” that revealed the sites where nascent microtubules emanated. We demonstrated that the centrosomes were equally active in both cell lines, whereas the nucleating activity of the Golgi was approximately three times higher in BS-C-1 cells. Taken together, these data indicate that the BS-C-1 cells possessed more “powerful” Golgi in terms of its microtubule-organizing activity, whereas the Golgi of the Vero cells was a “weak” MTOC.

To obtain a better understanding of the mechanisms underlying the Golgi activity as an MTOC, we attempted to enhance it in the Vero cells. Since, among the MT and Golgi-associated proteins, only CAMSAP3 was expressed differentially in the studied cell lines, we studied it first. Our nocodazole washout experiments, in accordance with previously published results [28,29], showed that non-radial growth of microtubules became dominant in transfected cells. Interestingly, according to another study, the moderate overexpression of CAMSAP3 actively reduced tubulin acetylation in both epithelial and neuronal cells [35]. CAMSAP3 thus seems to serve the purpose of retaining a pool of dynamic microtubules in axons, although their microtubules are generally acetylated. In any case, no visible colocalization between multiple cytoplasmic microtubules and dispersed Golgi vesicles was found in CAMSAP3-expressing Vero cells.

Previous studies identified several key participants in microtubule organization at the Golgi. Among them, the CLASP2 protein, originally discovered as a plus-tip stabilizer [36], was of greater interest as it was shown to be an essential regulator of microtubule organization at the Golgi [4]. However, it seems obvious that such a complex process cannot be attributed to a single protein or its direct partner(s). Moreover, depletion/rescue experiments of certain proteins sometimes make it impossible to provide a clear picture of the process and list all its participants. Our next attempts to increase the Golgi microtubule-organizing activity via the expression of CLASP2 or its adapter GCC185 also failed. Therefore, we could conclude that the GCC185-CLASP2 axis did not determine the difference in the Golgi properties between the two cell lines. This gave us a reason to assume the existence of some unknown participants in the process of microtubule organization at the Golgi membranes. 

Analysis of the transcriptome revealed similar expression levels for most of the known participants of MT organization onto the Golgi. Among potential candidates, one can point out the Janus kinase and microtubule-interacting proteins 2 and 3 (JAKMIP2/3) and GTPase Arl4C. JAKMIP (Janus Kinase and Microtubule Interacting Protein) is a family of three genes encoding multiple isoforms of long coiled-coil proteins [37]. JAKMIP1 (known also as Marlin-1 and GABABRBP [38]) participates in the regulation of the traffic of secretory vesicles in neuroendocrine cells [39]. It was shown that Marlin-1 binds to microtubules and Golgi membranes in neurons [40]. However, almost nothing is known about JAKMIP2 (JAMIP2) and JAKMIP3 (JAMIP3, NECC-2) functions [41]. Small GTPase Arl4c is also a protein with unknown functions, though a member of the same family, Arl4a, interacts with GCC185 in a GTP-dependent manner and is needed for the structural integrity of the Golgi [42]. 

Interestingly, the dominance of the centrosome as the main MTOC in the Vero cells correlates with a high level of expression of the second isoform of gamma-tubulin (TUBG2), though this isoform usually presents only in embryos and in the brain [32]. Since embryonic cells usually pass through the cell cycle much quicker than the cells from adult tissues, their centrosomes might be prone to holding their role as the only MTOC during the relatively short interphase period. It was also shown that γ-tubulin-2 could accumulate during oxidative stress in mature neurons, where the role of non-centrosomal Golgi-derived microtubules is well known.

All this suggests that complex processes such as microtubule organization cannot be easily reduced to the functioning of a few proteins. Differential expression of many genes, including the undiscovered ones, may define the state and activity of each of the MTOC. A complex interplay between the centrosome and the Golgi is still not well understood. A detailed examination of several proteins, which, according to our study, are possibly involved in the microtubule organization at the Golgi membranes, is yet to come.

## 4. Materials and Methods

### 4.1. Cultured Cells, Transfection, Inhibitors, Microtubule Regrowth, Cytoplast Preparation, Fixation 

Cultured green monkey kidney Vero (ATCC—CCL-81) and BS-C-1 cells (ATCC—CCL-26) were taken from the lab stocks. All cells were cultured in DMEM/F12 (1:1) medium (Paneco, Moscow, Russia) supplemented with 7.5% fetal calf serum (Paneco), glutamine, and penicillin/streptomycin, at 37 °C and 5% CO_2_. The transit-LT1 reagent (Mirus Bio, Madison, WI, USA) was used for transfection according to the manufacturer’s instructions. For Golgi dispersion, 3 μg/mL of brefeldin A (Sigma-Aldrich, Darmstadt, Germany) was added for 2 h. For centrosome depletion, the Plk 4 inhibitor centrinone B (Tocris, cat.# 5690) was added to the culture medium at a 500 nM final concentration, and the cells were fixed daily and immunostained to visualize the centrosomes. After 7 days of incubation with centrinone B, when no centrosomes were found, the analysis of microtubule morphology was performed. For microtubule depolymerization, 3 μg/mL nocodazole was added to the culture medium and, additionally, the cells were placed for 2 h at +4 °C. To follow microtubule regrowth, cells were briefly washed in the cold medium (+4 °C) and then transferred to the warm medium (+37 °C) for the required amount of time (2–15 min). The cytoplasts were obtained as described previously [43]. Briefly, cells were treated with nocodazole (1 μg/mL) and cytochalasin B (2 μg/mL) for 1.5 h. Coverslips were placed upside down into centrifuge tubes and centrifuged in a pre-heated Eppendorf 5414 centrifuge at 14,000 rpm for 10 min. Enucleation resulted in approximately equal numbers of cytoplasts containing or lacking the centrosome. Coverslips were washed with fresh medium and incubated at 37 °C for 2 h. Cytoplasts were then fixed and proceeded to immunofluorescence. 

### 4.2. Antibodies

The following antibodies were used for immunostaining: Mouse monoclonal to α-tubulin DM-1A (T9026, Sigma-Aldrich, Darmstadt, Germany), rat monoclonal YOL1/34 to tubulin (Ab6161, Abcam, Cambridge, UK), mouse monoclonal to p150^Glued^ (BD Transduction, cat. no. 610474), rabbit polyclonal to mannosidase II (12277, Abcam, Cambridge, UK), rabbit polyclonal to CLASP2 (Abcam, Cambridge, UK), rabbit polyclonal anti-GM130 (NBP2-53420, Novus Biologicals, Minneapolis, MN, USA), mouse Anti-GMAP-210 (BD Transduction, cat.no. 611712), rabbit polyclonal to γ-tubulin kindly provided by Dr. R. Uzbekov and also ab11321 from Abcam (Cambridge, UK). Species-specific anti-Ig antibodies (MultiLabeling class) conjugated with fluorochromes (FITC, TRITC, Cy5) were from Jackson ImmunoResearch Laboratories (Bar-harbor, ME, USA). Species-specific anti-Ig conjugated with alkaline phosphatase (KPL, USA) as secondary antibodies for immunoblotting were used. 5-Bromo-4-chloro-3-indolyl phosphate/nitro blue tetrazolium solution (KPL) was used for the staining development.

### 4.3. Immunofluorescent Staining and Microscopy, Live Imaging and Data Analysis 

Cells were briefly washed with PBS and fixed. Standard fixation protocol comprised two steps: Pre-fixation in absolute methanol for 7 min at −20 °C followed by subsequent fixation with 3% paraformaldehyde for 20 min at 4 °C. For immunostaining, primary antibodies were used at a concentration of 1–5 μg/mL, which normally corresponds to a dilution of 1:200 from their initial commercial concentrations. Secondary antibodies were used at the concentration of 5 μg/mL. The incubation time with both primary and secondary antibodies was 1 h at room temperature in the dark in a wet chamber. Coverslips were mounted in Aqua PolyMount (Polysciences, PA, USA)

Images were acquired using the Carl Zeiss Axiovert 200 M microscope supplied with 12-bit Axio-CamHR and 12-bit AxioCamMRc cameras and AxioVision software (Carl Zeiss, Jena, Germany). For live microscopy, cells were maintained at 37 °C. The temperature was maintained by the heating unit Tempcontrol 37-2 Digital (PeCon GmbH, Bielefeld, Germany) supplied with an incubation system for live observation. Alternatively, an inverted Olympus IX71 was used accompanied by a 14-bit CCD-camera Olympus XM10 and shutter UniBlitz D122 under the control of Micro-Manager software. Cells with a relatively low level of overexpression were selected for time-lapse microscopy. During live cell imaging, frames were acquired with an interval of 2 s. Frame-by-frame viewing of the image stack allowed us to manually mark and calculate the EB3 comets originating on the centrosome, since the centrosome was also clearly seen as a bright dot. The appearance of an EB3 comet from the point marking the centrosome was scored as a microtubule nucleation event. The average number of events per minute was analyzed as described previously [44]. To analyze MT nucleation on the Golgi during nocodazole washout, the total number of short microtubules that arose in the cytoplasm was counted in each of the analyzed cells. Then it was found how many of them co-localized with GM130-positive vesicles. Thus, for each of the 25 Vero and 25 BS-C-1 cells, the percentage of cytoplasmic microtubules extending from the Golgi vesicles was calculated from the total number of microtubules in the cytoplasm. The data were analyzed using ImageJ (National Institutes of Health, Bethesda, MD, USA), Excel (Microsoft, Redmond, WA, USA) and Origin 6.1 (OriginLab Corporation, Northampton, MA, USA) software.

For confocal imaging, a Zeiss LSM900 confocal microscope was used (provided by the Moscow State University Development Program). 

### 4.4. DNA Constructs

mCherry-GCC185: To obtain human GCC185 (Accession # NM_181453.3) cDNA, total RNA was isolated from HeLa cells using the RNeasy kit (74104, Qiagen, Hilden, Germany). First-strand cDNA was synthesized with Revert Aid reverse transcriptase (EP0441, ThermoFisher scientific) and random hexanucleotide primers. GCC185 cDNAs were amplified by parts, and internal oligonucletide primers hold endogenous restriction sites for KpnI, PciI, EcoRI restriction endonucleases (forward—TGGAGGATCTTGTTCAAGATGGGGTGGCT, GAAAAGTTACTATCTCAACAAGAATTGGTACCAGA, TAATTCAAGTTGAAGAAGTATCTCAAACATGTAGC, AGCTACTGTAACCTCTGAATTCGAGAGCTACA), (reverse—TCTGGTACCAATTCTTGTTGAGATAGTAACTTTTC, GCTACATGTTTGAGATACTTCTTCAACTTGAATTA, TGTAGCTCTCGAATTCAGAGGTTACAGTAGCT, CTATCGAAGTCCAGACCAACTATGAAGATAGG). The cDNA parts were verified by automated DNA sequencing and attached together using pairwise restriction, ligation, and amplification of the ligation product. The complete GCC185 cDNA were additionally amplified with BamHI and SalI addition primers (forward—ATAGGATCCATGGAGGATCTTGTTCAAGAT, reverse - ATAGTCGACCTATCGAAGTCCAGACCAAC) and cloned into the mCherry-C1 (Clontech, Mountain View, CA, USA) vector using these sites. GFP-CLASP2-GRIP was obtained by cloning the GRIP domain of Golgin-97 (described in [1]) in the frame before the natural stop codon of CLASP2 at the Smal restriction site. GFP-CLASP2 was a gift from Dr. A. Akhmanova and EB3-GFP was a gift from Dr. V. Rodionov. CAMSAP3-GFP plasmid (described in [28]) was kindly provided by Dr. A. Baffet.

All plasmids were purified using the MiniPrep kit (Evrogen, Moscow, Russia). All primers for molecular cloning were created in Synthol (Moscow, Russia). DNA constructs were verified by automated sequencing in the Genome Center (Moscow, Russia); the size of the protein expression products was confirmed by immunoblotting.

### 4.5. SDS-PAGE and Immunoblotting

Cells were lysed in 2× Laemmli sample buffer (65.8 mM Tris-HCl, pH = 6.8, 26.3% (*w/v*) Glycerol, 2.1% SDS, 0.01% bromophenol blue). SDS-PAGE was carried out in a BIO-RAD Mini-PROTEAN Tetra System using 10% homemade polyacrylamide gel in Tris-glycine buffer (0.25 M Tris base, 1.92 M glylcine) supplemented with 0.1% SDS. Wet transfer of proteins from a polyacrylamide gel to a nitrocellulose membrane (BioRad, CA, USA) was performed in a chamber filled with cold (+4 °C) Tris-Glycine buffer. The process was carried out at 99 V for 1–2 h. Membranes were blocked in the TBS buffer containing 0.05% Tween-20 and 5% skim milk and incubated sequentially in the solution of the primary and secondary antibodies. Anti GMAP210 and CLASP2 described above were used 10-fold diluted in comparison with their dilution for the immunofluorescent staining procedure. 

### 4.6. RNA Sequencing

Total RNA was extracted using the RNeasy kit (Qiagen, Hilden, Germany) following the manufacturer’s protocol. Illumina cDNA libraries were constructed with the TruSeq RNA Sample Prep Kits v2 (Illumina, San Diego, CA, USA) following the manufacturer’s protocol. Sequencing of the cDNA libraries was performed using an Illumina HiSeq2000. Reads were trimmed using the CLC Genomics Workbench 9.4.1 with the following parameters: ‘qualityscores—0.005; trim ambiguous nucleotides—2; remove 5′ terminal nucleotides—1; remove 3′ terminal nucleotides—1; discard reads below length 25’. Trimmed reads were mapped using the RNA-seq mapping algorithm implemented in the CLC Genomics Workbench to the reference Chlorocebus_sabeus genome V 1.1. allowing only unique mapping (length fraction = 1, similarity fraction = 0.95). Differentially expressed genes were inferred using the R package “DESeq2”. FDR < 0.05 and 0.5 < FC < 2 were used as thresholds of statistical significance. 

## Figures and Tables

**Figure 1 ijms-23-16178-f001:**
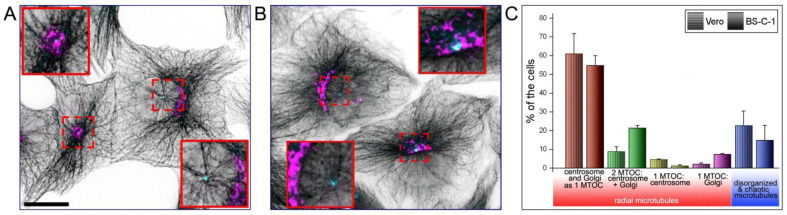
Microtubule arrangement in control Vero and BS-C-1 cells. (**A**) Triple immunostaining of Vero cells with antibodies against α-tubulin (black, inverted contrast), mannosidase II as a Golgi marker (magenta), and p150^Glued^ as a centrosome marker (cyan). The cell on the left demonstrates the radial array of microtubules diverging from the single MTOC, which is formed by both the centrosome and the Golgi. In the right cell, the second MTOC formed by the Golgi is placed apart from the centrosome. The outlined areas are shown enlarged in red squares. Scale bar, 15 μm. (**B**) Similar panel with BS-C-1 cells. (**C**) Analysis of MT organization in Vero and BS-C-1 cells. The number of analyzed cells is 378 for Vero and 428 for BS-C-1 in 2 independent experiments. Most cells of both lines demonstrate the radial aster of microtubules, whereas disorganized or completely chaotic microtubules were observed in only 22.2% of Vero and in 16.6% of BS-C-1 cells. When the Golgi and centrosome are spatially separated, two distinguishable centers of organization could be clearly observed (21.3% in BS-C-1 and 8.7% in Vero). Rarely, microtubules extend from the centrosome only, but not from the Golgi (1.2% in BS-C-1 and 4.8% in Vero) and vice versa, while sometimes microtubules extend from the Golgi only (7.2% in BS-C-1 and 2.1% in Vero).

**Figure 2 ijms-23-16178-f002:**
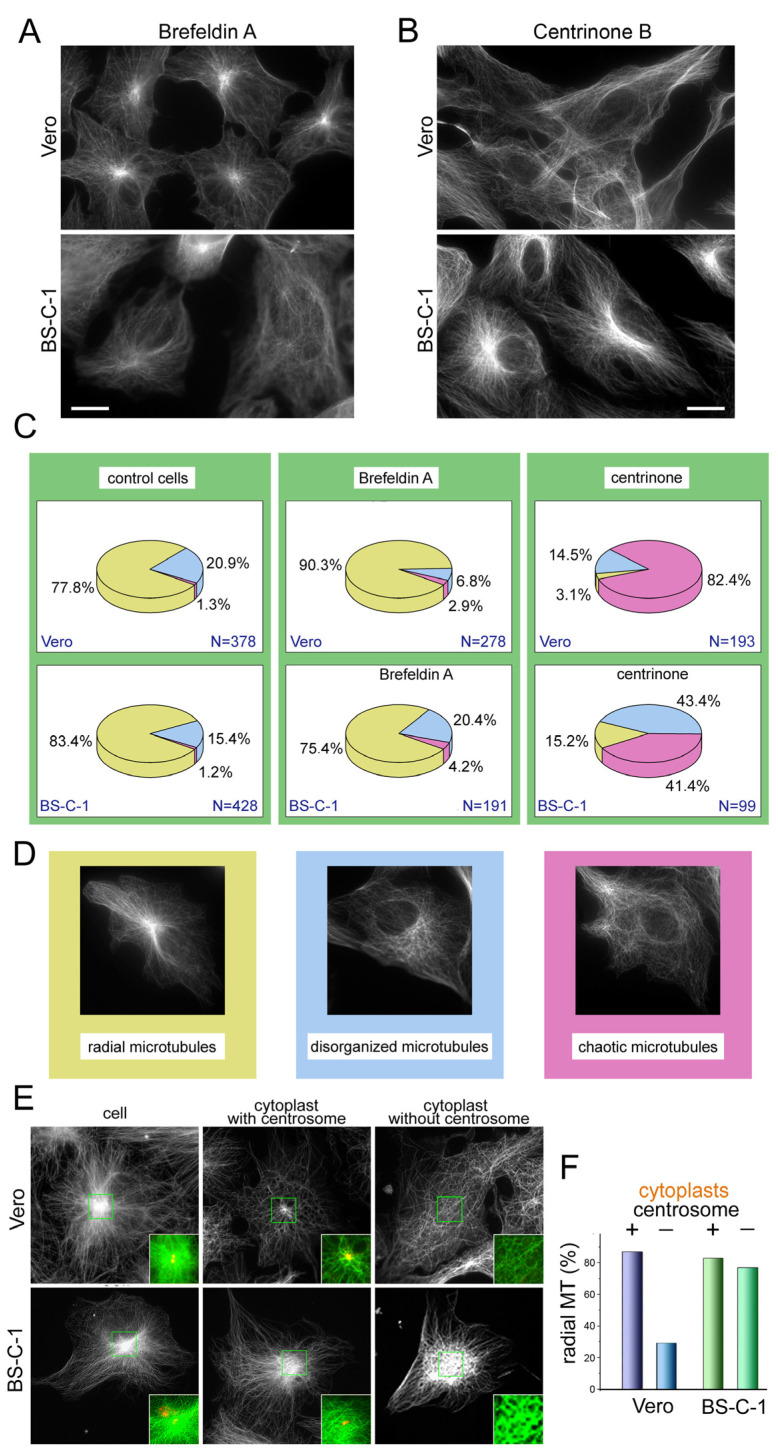
Disruption of the Golgi or the centrosome affects microtubule organization differently in BS-C-1 and Vero cells. (**A**) Immunostaining of Vero and BS-C-1 cells with antibodies against α-tubulin demonstrate the microtubule architecture after Brefeldin A treatment, which causes total disruption of Golgi ribbon (mannosidase II staining is not shown). Bar, 20 μm. For multiple staining see Appendix A. (**B**) Immunostaining of Vero and BS-C-1 cells with antibodies against α-tubulin demonstrate microtubule architecture after centrinone treatment, which causes loss of the centrosomes in the cells (γ-tubulin staining is not shown). Bar, 20 μm. For multiple staining see Appendix A. (**C**) Analysis of changes in MT organization in Vero and BS-C-1 after Golgi disruption by Brefeldin A or centrosome abolishing by centrinone B. Total number of analyzed cells is indicated near the pie plots as N. Different colors of sectors correspond to the types of microtubule organization—radial, partially disorganized, or completely chaotic. (**D**) Types of microtubule organization in the cells. (**E**) Microtubules in BS-C-1 and Vero cells and cytoplasts, containing or lacking the centrosomes. Immunostaining with antibodies against α-tubulin (white or green in enlarged squares) and γ-tubulin (yellow in enlarged squares). Vero cytoplast without the centrosome demonstrates completely chaotic microtubules. (**F**) Analysis of MT radiality in BS-C-1 and Vero cytoplasts, containing or lacking the centrosomes. MTs were analyzed in 23 centrosomal and 20 non-centrosomal Vero cytoplasts and 23 centrosomal and 21 non-centrosomal BS-C-1 cytoplasts.

**Figure 3 ijms-23-16178-f003:**
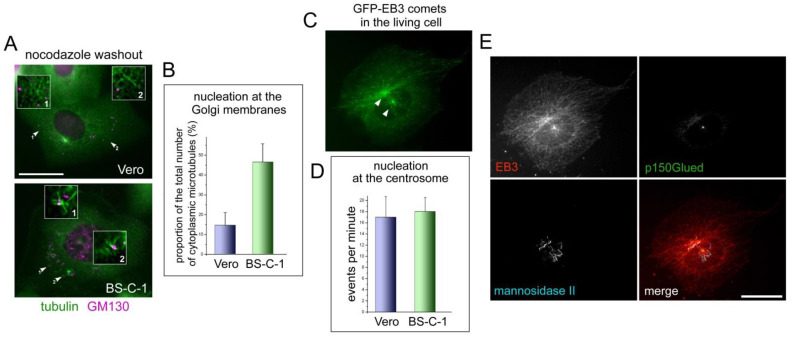
Microtubule-nucleating activity of the centrosome and the Golgi in BS-C-1 and in Vero cells. (**A**) Double immunostaining of Vero and BS-C-1 cells with antibodies against tubulin (green) and GM130 (magenta) after 4 min of nocodazole washout. Colocalization or non-colocalization of short cytoplasmic microtubules and GM130-positive vesicles in the cells are marked by the arrows and shown enlarged. Bar, 20 μm. (**B**) Analysis of colocalization of cytoplasmic microtubules and GM130-positive vesicles in nocodazole washout experiments. Twenty-five cells of each cell line were analyzed. Approximately three times more microtubules were associated with GM-130-positive dots in BS-C-1 cells than in Vero (46.55% and 14.68% correspondingly). (**C**) Single frame from Movie S1, demonstrating the GFP-EB3 comets emanating from the points of microtubules’ nucleation in BS-C-1 cell. (**D**) Analysis of MT nucleation at the centrosomes: Number of GFP-EB3 comets originating from the centrosomes per minute was counted. Eleven videos with Vero cells and nine with BS-C-1 cells were analyzed. (**E**) The same cell shown in (**C**) after fixation and immunostaining with antibodies against mannosidaze II (cyan) and p150^Glued^ (green). Red and green pseudocolors were exchanged in this panel for a better view. Bar, 20 μm.

**Figure 4 ijms-23-16178-f004:**
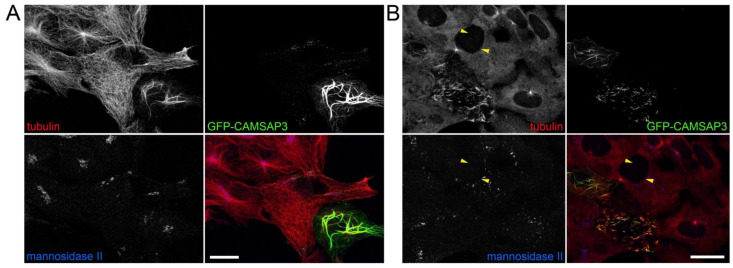
CAMSAP3 affects the non-centrosomal nucleation of microtubules in Vero cells in a Golgi-independent manner. (**A**) Vero cells overexpressing CAMSAP3-GFP; maximum projections obtained from Z stack. The cells were immunostained with antibodies against tubulin (red) and mannosidase II (blue). Microtubule randomization or bundling depending on the level of overexpression is shown. The Golgi ribbon is fragmented in both transfected cells. Bar, 20 μm. (**B**) Nocodazole washout in Vero cells overexpressing CAMSAP3-GFP; maximum projections obtained from Z stack. Numerous non-centrosomal microtubules fill the cytoplasm of transfected cells without any correlation with Golgi vesicles’ location. The same free cytoplasmic non-centrosomal microtubules in control cells are shown by arrows. Bar, 20 μm.

**Figure 5 ijms-23-16178-f005:**
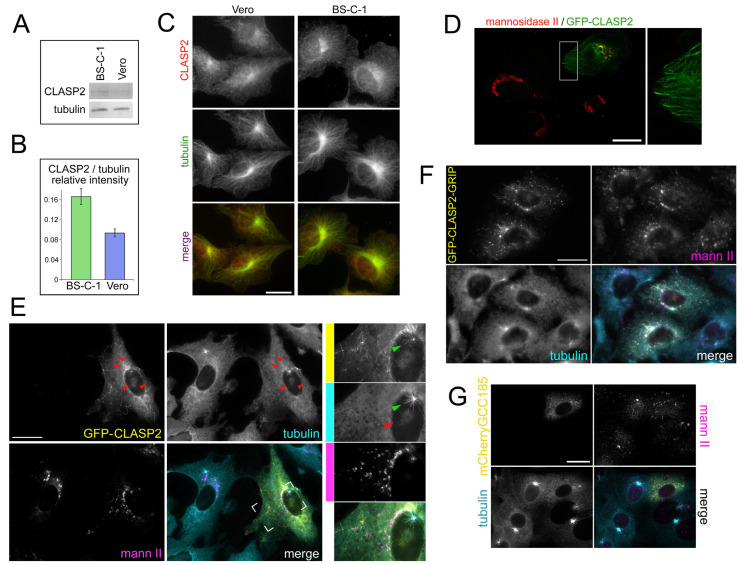
CLASP2/GCC185 - dependent mechanism does not determine the difference in Golgi microtubule-organizing activity among BS-C-1 and Vero cells. (**A**) Immunoblotting of BS-C-1 and Vero cell homogenates with anti-CLASP2 and anti-tubulin antibodies. CLASP2 level is slightly elevated in BS-C-1 cells. (**B**) Analysis of three WBs as shown in panel (**A**): Relative intensity of CLASP2/tubulin. (**C**) Double immunostaining of Vero and BS-C-1 cells with antibodies against α-tubulin (green) and CLASP2 (red). Localization of endogenous CLASP2 is similar in Vero and BS-C-1: It is distributed throughout the cytoplasm, having some affinity for the central region. Bar, 20 μm. (**D**) GFP-CLASP2 (green) overexpressing in Vero cell, stained with the antibodies to mannosidase II (red). GFP-CLASP2 localization along the MT is demonstrated in enlarged region. Bar, 20 μm. (**E**) Nocodazole washout in Vero cells expressed GFP-CLASP2 (yellow). The cells were stained with antibodies against α-tubulin (cyan) and mannosidase II (magenta). Cytoplasmic MT decorated by GFP-CLASP2 is marked with red arrows. The outlined area is shown enlarged on the right. Centrosomal astral microtubules decorated by GFP-CLASP2 are marked with green arrows. Bar, 20 μm. (**F**) Microtubule regrowth in Vero cells expressing chimeric construct GFP-CLASP-GRIP, which targets fused proteins to the surface of the Golgi (yellow). The cells were stained with antibodies against α-tubulin (cyan) and mannosidase II (magenta). GFP-CLASP-GRIP colocalizes to dispersed Golgi vesicles, but large proportion of cytoplasmic microtubules decorated with CLASP2 do not colocalize with mannosidase II—positive dots. Bar, 20 μm. (**G**) Microtubule regrowth in Vero cells expressing mCherry-GCC185 (yellow). The cells were stained with antibodies against α-tubulin (cyan) and mannosidase II (magenta). Exogenous GCC185 decorated numerous vesicles, which partially coincided with mannosidase II staining. GCC185-expressing Vero cells demonstrate decreased levels of microtubule nucleation both in the cytoplasm and at the centrosome. Bar, 20 μm.

**Table 1 ijms-23-16178-t001:** Genes related to microtubules and Golgi: Expression levels in BS-C-1 and Vero cells.

Name	Alternative Name	BS-C-1	Vero	BS-C-1/Vero
		Reads	Reads	
AKAP9	AKAP450	1754	1871	0.94
CAMSAP1		1215	1737	0.7
CAMSAP2		2338	1909	1.22
CAMSAP3		200	54	3.75
CDK5RAP1		574	866	0.66
CDK5RAP2		747	880	0.85
CDK5RAP3		1194	1669	0.72
CEP350		1584	1552	1.02
CLASP1		1224	1038	1.18
CLASP2		1276	1338	0.95
CLASRP		300	530	0.57
GCC2	GCC185	1529	1056	1.45
GOLGA2	GM130	1181	989	1.19
GORASP1	GRASP65	1678	940	1.79
MMG *	LOC103247162	0	0	0
	LOC103224073	5	2	2.5
	LOC103224075	0.5	0.5	1
MTCL1	CCDC165	1413	871	1.62
NIN	ninein	1610	1539	1.05
TRIP11	GMAP210	1062	1366	0.78

* MMG closely related protein PDE4DIP was not found in the transcriptome list.

## Data Availability

The Illumina sequence reads have been deposited into the NCBI Sequence Read Archive under project id PRJNA862476.

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
