# Peer review of "Divergent Contribution of the Golgi Apparatus to Microtubule Organization in Related Cell Lines"

_ijms, 2022, doi:10.3390/ijms232416178_

Round 1

Reviewer 1 Report (Previous Reviewer 3)

The authors have done a good job with the revisions and there are no concerns with the final manuscript.

Author Response

>The authors have done a good job with the revisions and there are no concerns with the final manuscript.

We express our gratitude to all the reviewers whose work allowed us to improve the quality of the manuscript. We have made some minor corrections to the text, mainly in the "Materials and Methods" section, to make it easier to read.

Reviewer 2 Report (New Reviewer)

In this manuscript entlited “Divergent contribution of the Golgi apparatus to microtubule organization in related cell lines” Brodsky and collegues studied the involvement of Golgi apparatus in microtubule organization in two cell lines, both originated from green monkey renal epithelium. The Authors observed that microtubules polymerization relied either on the centrosome or on the Golgi as a main microtubule-organizing center. Moreover, Brodsky and collegues reported a difference in the two cellular lines of Golgi microtubule-organizing activity not associated with the well-studied proteins, but with other several potential candidates involved in this process. This is a very interesting paper since recent data revealed the role of a special subset of Golgi-derived microtubules, involved in vesicular intracellular traffic. The Authors employed several different experimental approaches, and the experimental work is well orchestrated.

The paper is surely of interest and worthy of being published, since membrane trafficking along microtubules (MTs) plays an essential role in a variety of intracellular processes.

However, this reviewer retains that some revisions are necessary to make it accessible to a wider audience of readers.

Here my specific comments:

Results

Page 6 lines 206-214

The main concern this reviewer is the method employed to quantify the microtubules nucleated at the centrosome. About this point this referee retains that the Authors should clarify how they counted the EB3 comets arising from centrosome. I mean, since EB3 marks plus-end of microtubules, the Authors should better explain how they exclude the possibility that they are considering different events of nucleation and not the same.

Page 8 line 303

“To answer could the level……” may be “To answer if it could be the level…”

Page 9 line 329

Here the Authors should report citations to support this claim.

Materials and Methods

It must keep in mind that Materials and methods must be carefully described to allow the reproducibility of the experiments.

Page 14 line 565

About centrinone B the Authors omitted to indicate the time of incubation. They should clarify.

Page 14, Antibodies section.

In this section the Authors do not indicate neither the antibodies concentrations employed neither the incubation times of cells in the different antibodies. The Authors must integrate these additional informations.

 Page 15 SDS-PAGE and immunoblotting section.

In this section the Authors do not well describe methods and times of both electrophoresis and electrotransferred procedures of proteins to nitrocellulose. The Authors must clarify this point. Moreover, the Authors must indicate antibodies concentrations as well as their incubation times.

Author Response

We express our gratitude to all the reviewers whose work allowed us to improve the quality of the submitted manuscript. We have made some minor corrections to the text, mainly in the "Materials and Methods" section, to make it easier to read. Below you can find our responses to specific comments.

>>Page 6 lines 206-214  Authors should better explain how they exclude the possibility that they are considering different events of nucleation and not the same.

We made the necessary changes in the sections "results" and "materials and methods"

>>Page 8 line 303  “To answer could the level……” may be “To answer if it could be the level…”

We have replaced the corresponding phrase in the text

>>Page 9 line 329 Here the Authors should report citations to support this claim.

We have given in this place the links to reviews in which these articles are analyzed.

>> Page 14 line 565 About centrinone B the Authors omitted to indicate the time of incubation. They should clarify.

We specified on which day of exposure the microtubules were analyzed. In addition, in the Figure 2 we replaced the word "centrinone" with "centrinone B"

>> Page 14, Antibodies section. In this section the Authors do not indicate neither the antibodies concentrations employed neither the incubation times of cells in the different antibodies. The Authors must integrate these additional informations.

We rewrite the corresponding section in "Materials and methods" and indicated the dilution of antibodies, as well as the time and conditions of incubation

 >>Page 15 SDS-PAGE and immunoblotting section. In this section the Authors do not well describe methods and times of both electrophoresis and electrotransferred procedures of proteins to nitrocellulose. The Authors must clarify this point. Moreover, the Authors must indicate antibodies concentrations as well as their incubation times.

We rewrite the corresponding section in "Materials and methods" and indicated the required parameters as well as the dilution of antibodies for Western blot

This manuscript is a resubmission of an earlier submission. The following is a list of the peer review reports and author responses from that submission.

Round 1

Reviewer 1 Report

The work represents a continuation of the research on specific microtubules associated with the Golgi apparatus. 

An important evidence of different microtubule organization in BS-C-1 and Vero cells lines is the effect of disruption of the Golgi or the centrosome affecting microtubules differently in cells demonstrated in the paper.

Additionally, after nocodazole washout the microtubule-nucleating activity of Golgi was indeed higher in BS-C-1 versus Vero, since it provided more microtubules nucleated not at the centrosome, but in association with the Golgi vesicles. 

It was demonstrated that the centrosomes were equally active in both cell lines, whereas nucleating activity of the Golgi was about three times higher in BS-C-1 cells.

Although the authors' attempt to increase Golgi microtubule-organizing activity by expression of CLASP2 or its adapter GCC185 did not succeed, the results of the transcriptome analysis performed by the authors allow us to outline new ways to study the role of the Golgi complex in the organization of the microtubule system in cells.

I have no significant comments on the results of this work, all conclusions are substantiated and are discussed in comparison with previously published results of other researchers on this topic.

It may make sense to use the abbreviation MT for microtubules, since that is already a common abbreviation.

Author Response

We thank this Reviewer who thinks that all conclusions are substantiated and are discussed in comparison with previously published results of other researchers on this topic.

Reviewer 1 had no significant comments on our results, so we made the changes to the text and figures in accordance with the comments of other reviewers.

Reviewer 2 Report

In this manuscript, Brodski et al. aimed to analyze a possible distinct relationship between two forms of microtubule nucleation, namely, from the centrosome or from the Golgi apparatus, among different types of cultured cells. Overall, the manuscript has merit, however, the data appear preliminary and do not warrant publication as it is.

Major issues:

2.1. Morphological analysis of the microtubule system revealed a difference in microtubule architecture between the two lines of green monkey cells

On the one hand, the morphological analysis of microtubule architecture comparing only two cell lines is not sufficient to draw any conclusion regarding possible differences in the relationship between the two forms of microtubule nucleation in different cells. This is an overall limitation of the manuscript. On the other hand, the morphological data presented in Figure-1 do not have the standard necessary to draw any conclusion. It is well known that the centrosome position and the overall architecture of the Golgi apparatus vary during the cell cycle, and this heterogeneity, which is present under the cell growth conditions used throughout the manuscript, is not taken into account in the study. Moreover, in populations of adherent cells, there is intrinsic heterogeneity in cell morphology due to different cellular states, and this correlates with different states of microtubule organization. In this regard, the study also does not offer an analysis that considers this additional level of heterogeneity. Regarding the presentation of the data, because the study is centered on possible morphological differences, it is recommended that figures include panels of microscopy images of each protein analyzed in addition to merged images. This is important for a better assessment of the possible morphological differences.

2.2. Centrosome and the Golgi have different impact to microtubule organization in Vero and BS-C-1 cells

Authors indicate that brefeldin A produces larger damage to microtubules in BS-C-1 cells than in Vero cells, however, they do not describe how was assessed the level of damage and how the level of structural change of microtubules was quantified. In addition, the absence of images of untreated (control) cells in Figure-2 and of images of the effect of brefeldin A on the Golgi preclude the assessment of an effect of brefeldin A on microtubules via the Golgi in BS-C-1 cells. Likewise, the authors indicate that brefeldin A in Vero cells generates more radial microtubules, however, they do not describe how this effect was quantified. Authors also used centrinone B to disrupt centrosomes and indicate an almost total lack of ordered microtubules in Vero cells, but again, they do not describe how this effect was quantified. 

2.3. Different nucleation activity at the Golgi membranes in Vero and BS-C-1 cells coexists with the same activity of their centrosomes

To assess the effect of nocodazole on microtubules and the Golgi in Vero and BS-C-1 cells, the nocodazole concentration used should have been higher, at least 10 ug/ml (33 uM) instead of 3 ug/ml, otherwise incomplete microtubule depolymerization occurs. On the other hand, in the methodology section, it is stated that nocodazole treatment was performed at 4 ºC. Because incubation of cells at 4 ºC causes depolymerization of microtubules, it is not clear why the combination of treatments (nocodazole and 4 ºC) was used. In Figure 3, regarding double staining to detect GM130 and tubulin, or mannosidase II and p150Glued in cells expressing GFP-EB3, unmerged images are necessary to assess the results. 

2.4. The observed difference in Golgi activity as a MTOC is not due to the previously described molecular mechanisms

In this section, the authors aimed to determine whether proteins known as microtubule nucleation and anchoring factors onto the Golgi were differentially expressed in Vero and BS-C-1 cells, assessed by transcriptomic analysis. They found no difference in the level of mRNA expression of a number of Golgi proteins. However, because the level of mRNA expression does not always correlate with protein levels, to draw any conclusion on this an immunoblot analysis should have been performed. On the other hand, to directly assess the involvement of some proteins in microtubule organization at the Golgi, the authors performed exogenous overexpression experiments. The authors found no effects on microtubule organization, however, they tested a limited group of proteins. Thus, this section appears incomplete, but the authors conclude that unknown factors should exist in Vero cells that allow microtubule nucleation at the Golgi to the level they found in BS-C-1 cells.

2.5. Transcriptomic analysis has revealed several hitherto unknown potential factors that may influence the Golgi functioning as a microtubule organizer 

In this section, the authors extended their search of microtubule nucleation factors to a global transcriptomic analysis. They found overall "similarity in the level of gene expression of almost all proteins", but at the same time they found "a number of significant differences ... in the level of expression of a number of genes ... that could contribute to these processes". However, they did not test whether this set of proteins is in fact important for microtubule nucleation. Overall, this section is largely preliminary and does not contribute to the overall aim of the manuscript. Moreover, the title of the manuscript is misleading, because this last analysis or any other throughout the manuscript reveals novel regulators of microtubule organization at the Golgi apparatus.

Minor issue:

Although the manuscript is generally easy to read and understand, further proofreading is recommended.

Reviewer 3 Report

This paper shows an interesting phenomenon regarding the Golgi ncMTOCs between two Green Monkey kidney epithelial cell lines: Vero cells and BS-C-1 cells. The authors show that the relative activities of the Golgi MTOC between the two cell lines is significantly different, with BS-C-1 cells having a more robust one compared to Vero cells (or Vero cells have a much attenuated one compared to BS-C-1 cells) and show that Vero cells rely more on MT organization from the centrosome. Another available line, the CV-1 cell line from green monkey kidney epithelium, could also have been compared. The authors used two approaches to discern differences between centrosome and Golgi MTOCs: treatment with brefeldin-A and centrinone in one approach, and the generation of cytoplasts in the other. Together the combined approaches rigorously show that BS-C-1 cells have a more robust Golgi MTOC compared to Vero cells.

The authors approached deciphering the differences between these two lines by RNAseq transcriptome analysis and by examining one candidate protein axis, CLASP2, and found no significant impact from this factor. The decision to focus on CLASP2 was unclear since its expression did not differ between the cell lines. The GCC185 overexpression results are interesting, where GCC185 overexpression depleted MT organization activity at the centrosome, but the interpretation is unclear. The transcriptomic analysis revealed significant differences in the expression profiles between the two cell lines despite their common origin.

 The major finding of the work is the demonstrable difference seen between the Golgi MTOCs of two cell lines. The authors did not find any factors that account for it, and suggest, based on the transcriptomics, that novel factors might be involved. The authors do not demonstrate this, and transcription is certainly not the only way that expression can be controlled, so the authors cannot rule out the involvement of established Golgi MTOC factors without a further survey of the proteins expressed from the candidate genes. The authors present an intriguing system to dissect the control of the Golgi MTOC and how it might differentially affect the secretion functions of related cell types. The study therefore nicely describes the differences in the Golgi MTOC between these two cell lines, and the extensive differences in their transcriptomes, but does not discern any mechanistic basis for the difference in the Golgi MTOCs. 

Overall, the work is therefore preliminary and does not significantly advance our understanding on the function of the Golgi MTOC or cell-type control over the proces, but nicely sets the groundwork for future investigations. Additionally, it is probably not appropriate for the title of the paper to conclude that novel regulators were revealed as none were identified. The paper may be more appropriate for a journal with a scope and audience that is not as broad as that of IJMS.

Specific points:

1.     The title of the paper should be revised as no specific factors were identified. In addition, the statement in the abstract that the difference between the Golgi MTOCs is not due to the well-studied proteins is also not well supported by the data as the proteins were not investigated. 

2.     In Figure 1, the authors should include a representative image of the BS-C-1 cells for comparison. Only the Vero cells are shown.

3.     For the quantitation of MT morphologies in cells after different drug treatments in Figure 2, perhaps the authors can explain the difference between the “disorganized” and “chaotic” categories. Some example images could be included in Figure 2 or in supplemental.

4.     In Figure 3 a magnified inset is provided to show MT regrowth at BS-C-1 Golgi but not at Vero Golgi, which should also be included. If both show similar regrowth, the difference in MTOCs could be due to differential MT anchoring ability.

5.     In the right panel of Figure 3C, the p150Glued and manosidase II signals cannot be discerned from the image. Please provide separate panels.

6.     Regarding the transcriptome analysis of MMG, the authors designate “N/D” next to PDE4DIP. This needs to be defined somewhere, perhaps in the footer of the table. Is it “Not Determined”, “No Difference”, “No Data” or something else? Also, for MMG, what does it mean for there to be 0.5 reads for LOC103224075 for each cell line? Perhaps it means there was 1 read in one replicate and zero in the other?

7.     The authors rule out CAMSAP3, which was one of the factors whose transcription levels appeared to change significantly, by stating that it has not been shown to control the Golgi MTOC. But has it also been shown not to? I could not find a paper that concluded that it did not play a role at the Golgi MTOC. And importantly it could be involved in cell-type specific ways. This could be evaluated by IF staining. The authors make the point that novel, perhaps unknown, factors might regulate the Golgi MTOC in BS-C-1 cells. In line with this thinking, such a factor could conceivably act through an established factor like CAMSAP3. Ultimately, a major stated goal of the transcriptomic analysis was to identify candidates that account for the relatively higher Golgi MTOC in BS-C-1 cells (or lower in Vero), and so it falls to reason to follow up on a ‘hit’ like CAMSAP3.

8.     The rationale for focusing on CLASP2 and GCC185 and proceeding with overexpressing GFP-CLASP2, GFP-CLASP2-GRIP, and mCherry-GCC185 is not clearly described. Since the overall conclusion is that there appears to be no role attributed to CLASP2 to account for the differences in Golgi MTOCs between Vero and BS-C-1 cells, perhaps these data (Figure 4) should belong in supplemental. After all, there was no clear premise since the transcriptomics showed no significant difference for CLASP2. Rather than overexpress CLASP2 in Vero cells, the authors could have knocked it down in BS-C-1 cells to assess its role in the differential Golgi MTOC activity between the two cell lines.

9.     Figure 4A: If the conclusion that CLASP2 protein is lower in Vero cells from this one blot, then the authors need to repeat it at least two more times and quantify the signal relative to the loading control.

10.  Figure 4C: it is not clear how this panel shows that overexpressed GFP-CLASP2 localized along microtubules as stated on lines 327-330. It is not indicated that microtubule staining was included in this panel and moreover this panel is not labeled like the other panels. It is also not clear that GFP-CLASP2 is decorating centrosomal astral microtubules in Figure 4D (lines 334-335). There are some filamentous structures proximal to the nuclear surface – is this what the authors are referring to? For Figures 4C and D please label consistently and add arrows to point out the features that are described (like microtubules decorated with GFP-CLASP)

11.  In the Discussion, the authors state that “Golgi depletion appears to be more deleterious for cells.”(lines 457-458). The basis for this statement/conclusion is unclear as there are no supporting citations or data from this paper to support it. The following sentence also seems out of place or without context and needs clarification or should be removed: “Disruption of crucial processes such as protein processing and transport may affect microtubules indirectly.”

12.  In the Discussion (lines 490-492) the authors comment on the relative transcript levels of established Golgi MTOC protein genes and conclude that “Therefore, the difference we observed in our experiments was not due to differential expression of these key proteins.” The authors should modify this conclusion since they did not assess protein levels beyond CLASP2, and they recognized elsewhere in the manuscript that there are other levels of regulation of protein expression besides transcription.

13.  Most of the paragraph that begins on line 487 of the discussion is highly speculative and can be removed. 

Additional items:

1.     Line 67: whichis

2.     The two sentences on lines 267-269 are unclear and should be rephrased.

3.     The BS-C-1 cell line is sometimes referred to in the manuscript as BSC-1. The naming should be consistent.

4.     On line 461 it is unclear what “latter” refers to.

5.     The paragraph that spans lines 460-466 does not make a point. It probably needs to be fused with the next paragraph and revised for clarity.

6.     The term “affine” (line 488) is probably not appropriate to describe how the two cell lines are related. Instead, ‘cells of apparently related origin’ or something like that is more appropriate.

Author Response

We are grateful to this Reviewer, thanks to whose comments we were able to notice many shortcomings of our manuscript and allowed us to improve the article. We made changes in the text of the article and in the figures in accordance with the comments.

Specific points:

  1. We changed the title of the article as no specific factors were identified, and changed the corresponding statement in the abstract.
  2. We added an additional panel 1B with a representative image of the BS-C-1 cells for comparison.
  3. To explain the difference between the “disorganized” and “chaotic” categories, we added three images of microtubules with different degrees of orderliness as a panel 2D. The colors on the 2D panel correspond to the colors of the corresponding segments on the graphs of the 2C panel 
  4. We included a magnified insets in Figure 3A, to show MT regrowth in Vero cells. To analyze MT nucleation on Golgi during nocodazole washout, the total number of short microtubules that arose in the cytoplasm was counted in each of the analyzed cells. Then it was found how many of them co-localized with GM130-positive vesicles – for example, 15 out of 64 or 62 out of 123. Thus, for each of the 25 Vero and 25 BS-C-1 cells from two independent experiments, the percentage of cytoplasmic microtubules extending from the Golgi vesicles was calculated from the total number of microtubules in the cytoplasm.
  5. We provided the additional panels as a Figure 3E to show p150Glued, EB3 and mannosidase 2 separately.
  6. We added corresponding explanation about MMG in the footer of the Table 1. Indeed, N/D means “No data”, and 0.5 in the Table 1 is the average between 0 and 1 read from different probes.
  7. We add panel 4B to the Figure 4 to show the analysis of relative intensity of CLASP2/tubulin from 3 blots
  8. We added labels on the panel 4D (former 4C) and also included a magnified inset. We also added green and red arrows on the panel 4E (former 4D) to show the astral and cytoplasmic MTs decorated with GFP-CLASP.

We have changed the text of the article significantly in accordance with comments 7,8, 11,12 and 13. We also took into account the additional items 1-6 and have changed the text accordingly. Also we included a number of recently published papers which we missed in initial version of manuscript. After that the manuscript has undergone English language editing by MDPI. The text has been checked for correct use of grammar and common technical terms, and edited to a level suitable for reporting research in a scholarly journal

Reviewer 4 Report

This study is aimed at a better understanding of the functions of the Golgi network in organizing the MT array and vesicular transport in two different but related cell lines. Golgi network is long known as regulating MT nucleation in different cells (with an emphasis on specialized cells). As for centrosomes, MT nucleation from the Golgi depends upon gamma tubulin and many proteins were already identified as regulating MT nucleation from both sites. Since gamma-tubulin concentration at the Golgi membrane and cytoplasm are similar, the authors wish to identify novel proteins involved in gamma tubulin dependent MT nucleation at Golgi membranes.

The question, per se, is interesting, but the rationale of the study is quite poor.

First, the authors analyse two different cell lines emanating from green monkey renal epithelia and propose that MT nucleation at Golgi membranes in BS-C-1 cells is much more effective than in Vero cells. From this conclusion, they perform RNA seq of the two exponentially growing cell lines. They find no major differences in transcripts for proteins known to be involved in the assembly of MTs from the Golgi. But nonetheless, a bit naively, overexpress CLASP2 (a plus tip protein also involved in gamma tubulin dependent MT nucleation at Golgi membranes) to try to promote MT nucleation at Golgi membranes in these cells. Using different attempts, they get no conclusive results. Going back to their RNA seq, they analyze more throroughly the results and identify a number of transcript subset differently expressed in the two cell lines. They propose that some of these proteins might be involved in regulation MT nucleation at the Golgi membranes.

My main concern to start with is that the presented analyses, of the two cells lines, leading to the conclusion that MT nucleation from the Golgi network is more efficient in BS-C-1 cells are not sufficiently meticulous. Many controls are missing. Indeed, and as detailed below, double or triple staining to show the effectiveness of the different cell treatments are not provided. Few cells (in numbers) are analyzed and there is no indication of how the Data are quantified and whether the results provided are statistically relevant… for example MT intensity at nucleating centers in Figure 3..).

It is difficult at this stage to be confident in the presented Data and I would suggest a more thorough experimental design to reach the conclusions that MT nucleation from the Golgi network is much more effective in BS-C-1 than Vero cells. Such a study, with an opening on the differences of the RNA seq between the two cell lines, would be interesting to the reader by itself

The strategy used to overexpress potential candidate gene products such as Clasp2 (which also does not present much difference in expression between the two cell lines) in Vero cells to “change” its capacity to nucleate MTs from the golgi network was unlikeky to work out.

The English should be edited as some sentences are quite unclear.

The bibliography in the introduction section misses a number of recent studies

Major points

Figure 2. About the disruption of Golgi apparatus and centriole depletion/ Triple staining with mannosidase II and gamma antibodies should be shown in parallel to the MTs (as in figure 1) to show the effectiveness of the treatments

How many independent experiments were done?? Analyses should be performed on a larger number of cells (47 cells for centrinone treatment in BS-C-1 is quite a small number)

Figure 2D: Overlay of the gamma tubulin and tubulin staining should be shown in the low magnification images and not only on the enlarged region. DAPI would be informative to show that we are really looking at cytoplasts. Number of cells analyzed and number of experiments should be given. How old are the cytoplasts?

As a global comment, there is no indication of how the quantifications were performed.

Figure 3A: Enlarged region of the GM130 dots should be shown in the Vero cells as well (as in the BS-C-1 cells). 3B: How were performed the quantifications? What exactly means the “proportion of the total number of cytoplasmic microtubules?  25 cells is not many.. how many independent experiments were done?

Figure 3C: Live and fixed Vero cells should be shown as well. It is impossible to see the blue Mannosidase and green P150 glued staining from the image of the fixed BS-C-1 cell. The panel should be divided with single staining shown in grey levels.  I suppose that the number of MTs emanating from the centrosomes per minute is calculated from the live EB3 comets. Are the EB3gfp cell lines, stable cell lines??  Again, how is performed the “statistical processing of the movies”

Figure 4:

The rationale of proposed experiments is not very clear.   

The authors indeed stated: “We suggested to transform the Golgi of Vero cells to an active MTOC. To assess expression of the Golgi proteins, we performed a transcriptomic analysis of the studied cultures”..   

Related to Figure 4, lanes 323..  “To define whether Golgi capacity as MTOC depends mainly on CLASP2 we decided to overexpress it in Vero cells in order to reawake Golgi microtubule organizing activity in these cells. Unexpectedly, GFP-fused CLASP2 overexpression in Vero cells resulted in most of the exogenous protein being distributed along the microtubules. 

 This approach was very unlikely to work as overexpression of MT binding proteins often leads to changes in their dynamic and bundling..

Bibliography: Quite a large number of papers, that would be worth to be cited, were recently published on MT nucleating activity from the golgi network.

English is a problem throughout the text: Some examples are below

Some « principal proteins » lane 22

Lanes 28-29: Further transcriptome analysis revealed a difference in the expression levels of a number of genes that could be involved to the microtubules organization by Golgi. 

Lane 37 the microtubule system is rearranged in order to meet the spe-37 

gamma tubulin gamma- TuRCs 

 often indicated as @. Perhaps a pdf building problem

lanes 73-74:  Although some essential proteins that provide growth and retention of MTs on Golgi are already identified, the striking difference in Golgi ability to organize MTs observed in different cell types suggests that some participants in this process remain unknown. “

Lanes 86-87. This taken together suggests, that compensatory mechanisms might distort the final picture observed under inhibition or hyper-activation of any protein involved in this process 

Gamma and micro (figure legends indicated as @ throughout the text

etc ………….

Author Response

We are grateful to this Reviewer, thanks to whose comments we were able to notice many shortcomings of our manuscript and allowed us to improve the article. We made changes in the text of the article and in the figures in accordance with the comments. Also we included a number of recently published papers which we missed in initial version of manuscript. After that the manuscript has undergone English language editing by MDPI. The text has been checked for correct use of grammar and common technical terms, and edited to a level suitable for reporting research in a scholarly journal

Figure 2.

We add Supplemental Figures 1 and 2 with multiple stainings in addition to Figure 2A and 2B, to show the effectiveness of the different treatments.

We set up an additional experiment with centrinone B, which allowed us to increase the total number of analyzed cells (Figure 1C, plot on the right).

To explain the difference between the “disorganized” and “chaotic” categories, we added three images of microtubules with different degrees of orderliness as a panel 2D. The colors on the 2D panel correspond to the colors of the corresponding segments on the graphs of the 2C panel 

We stated in the Materials and methods that the age of cytoplasts was 2 hours after the enucleating, and the number of counted cytoplasts (from 20 to 23 for each column). We did not stain them with DAPI, since the presence or absence of a nucleus was clearly visible on phase contrast. We also added the pictures of the cells with nuclei to panel 2E.

Figure 3:

We included a magnified insets in Figure 3A, demonstrating short microtubules which are regrow in Vero cells and not associated with dispersed Golgi vesicles 

To analyze MT nucleation on Golgi vesicles in experiments with nocodazole washout, the total number of short microtubules that arose in the cytoplasm was counted in each of the analyzed cells. Then it was found how many of them co-localized with GM130-positive vesicles; for example, 15 out of 64 or 62 out of 123. Thus, for each of the 25 Vero and 25  BS-C-1 cells from two independent experiments, the percentage of cytoplasmic microtubules extending from the Golgi vesicles was calculated from the total number of microtubules in the cytoplasm.

We have presented in Figure 3E the multiple staining of a fixed cell as separate panels in gray levels.

The transient transfection of cells by GFP-EB3 encoding plasmid was used for imaging. Cells with a relatively low level of overexpression were selected for time-lapse microscopy. Frame-by-frame viewing of the image stack in ImageJ allow to manually mark and calculate the EB3 comets originating on the centrosome, since the centrosome was also clearly seen as a bright dot.

Round 2

Reviewer 3 Report

In the title, the word “various” could be misleading. Perhaps “divergent” or something similar would more accurately convey the dichotomous difference between the two cell lines.

I think the red arrows are missing in the insets for the revised Figure 4E. They are in the lower magnification panels, but readers cannot see what those arrows are indicating at that resolution.

Reviewer 4 Report

I rejected the paper in first instance because, there were too many concerns about the way the experiments were performed, about the accuracy of the quantifications, about the complete absence of statistical analyses.

The authors here provide a new version of their manuscript  in which they thoroughly edited the text.  I however still have the same concerns regarding the quantifications of the results.  I will just give an example here: In figure 3, it is very difficult to distinguish the small microtubules that regrow over the tubulin background (what about thresholding?). Nonetheless, the authors say they counted (manually, I understand) the short microtubules. It appears to me that it is impossible to do this just by eye and to be confident with the obtained results. It is even more challenging to be sure of potential colocalization of these short microtubules with GM130 positive dots as widefield microscopy is used.

In summary, this study should be strongly strengthen to confirm the conclusions drawn.